# Physical Exercise and Life Satisfaction of Urban Residents in China

**DOI:** 10.3390/bs14060494

**Published:** 2024-06-12

**Authors:** Buerzhasala Ha, Jie Zhang

**Affiliations:** 1School of Ethnology and Sociology, Minzu University of China, Beijing 100081, China; harper00@sina.com; 2School of Public Health, Shandong University, Jinan 250012, China; 3Department of Sociology, State University of New York Buffalo State University, Buffalo, NY 14222, USA

**Keywords:** life satisfaction, physical exercise, physical health, mental health, social communication

## Abstract

Currently, an increasing number of Chinese urban citizens are participating in daily physical exercise. Existing research has shown that physical exercise can increase life satisfaction. However, some studies also suggest that the relationship between the two is unstable. The purposes of this study are to examine physical exercise and to test its correlation with life satisfaction of urban residents in China. Data are obtained from the 2018 China Family Panel Studies, and we focus on urban residents. Our overall sample size is 7423 people, including 3641 females (49.05%) and 3782 males (50.95%), with an average age of 49.55 years old. Because the dependent variables are continuous variables, the multiple linear regression model is used for data analysis. We find that the life satisfaction of Chinese urban residents is on the high side. Our core discovery is that there is a significant positive relationship between the frequency and duration of physical exercise and life satisfaction. Our further discovery is that the frequency of physical exercise affects life satisfaction by influencing popularity and positive emotions. Similarly, the duration of physical exercise affects life satisfaction by influencing popularity and positive emotions. Whether it is the frequency or the duration of physical exercise, it can reflect the residents’ attention to physical exercise. Physical exercise habits not only promote physical health by strengthening physical fitness but also promote mental health by alleviating depression and promote social communication by increasing social activities in the Chinese context. All of these can improve people’s life satisfaction. Our research suggests that the improvement in life satisfaction not only needs the abundance of external material conditions but also needs the individual to improve their physical and mental health through physical exercise.

## 1. Introduction

Aristotle believed that happiness is the ultimate goal pursued by people, transcending any boundaries of material wealth and rights, and is the greatest value and meaning of life [1]. Life satisfaction refers to an individual’s cognitive assessment of his or her quality of life, which is a key indicator to measure his or her subjective well-being [2]. With the emergence of the Easterlin paradox [3], people’s understanding of life satisfaction has become more diverse. The Easterlin paradox shows that although the rich are usually happier than the poor within a country, when compared across borders, the happiness levels of poor and rich countries are similar, such as the United States and Cuba. The Easterlin paradox draws attention to the limitations of equating life satisfaction solely with material wealth. It emphasizes the importance of non-material factors such as health [4,5], psychological capital [6], and social relationships [7,8], encouraging people to have a broader understanding of life satisfaction beyond economic indicators.

Existing studies have shown that physical exercise is not only beneficial for physical health [9,10], such as preventing and treating chronic diseases [11], delaying aging [12], improving functional capacity, improving body composition [13,14], reducing risks of cardiovascular disease, obesity, stroke, and cancer [15], but it is also beneficial for mental health [16]. There is already a lot of research on the impact on physical health, so there is no need to elaborate further. The evidence regarding mental health comes from both neuroscience and psychology (The research content of this article is physical exercise rather than physical activity. According to Caspersen et al. [17], physical activity is defined as any bodily movement produced by skeletal muscles that results in energy expenditure. The energy expenditure can be measured in kilocalories. Physical activity in daily life can be categorized into occupational, sports, conditioning, household, or other activities. Exercise is a subset of physical activity that is planned, structured, and repetitive and has as a final or an intermediate objective the improvement or maintenance of physical fitness. However, the distinction between physical activity and physical exercise is not very clear in the existing literature, and the concept of physical activity is often used to measure the content of physical exercise. If the references are limited to the literature with the keyword physical exercise, a large number of valuable literature publications will be missed. Therefore, based on the specific content of the literature, the references in this article will also include the literature with physical activity as the keyword. This article is collectively referred to as physical exercise). 

Studies from neuroscience have shown that physical exercise can induce changes in brain structure and function [10], such as increasing gray matter volume in the frontal lobe and hippocampus, reducing gray matter damage, and expanding cortical and basal ganglia volume [10,18,19,20].

Studies from psychology have shown that physical exercise can reduce psychological disorders such as depression and anxiety [21,22,23], alcoholism [24], and even the probability of self-injurious behaviors [25]; improve emotional state [26]; and enhance self-esteem [27,28] and self-efficacy [29,30]. It should be noted that some studies have found that although overall exercise is better than not exercising, more exercise was not always associated with better mental health. If the duration exceeds 45 min per session or the frequency exceeds 5 times a week, the mental health burden of participants will increase [31]. There are also studies that have found gender differences in the impact of physical exercise on mental health. Boys and girls have different responses to physical exercise in terms of a reduced risk of depression, anxiety, ADHD, and ODD. For example, for boys, a high frequency of MVPA (Moderate-to-Vigorous Physical Activity) is significantly associated with a lower risk of depression. For girls, both moderate- and high-frequency MVPA are associated with a lower risk of depression, indicating a generally positive impact of physical exercise on the risk of depression in girls [32].

Physical exercise is a social behavior [33]. It can improve self-image and social skills [24]. Teenagers who exercise regularly become more sociable, which makes them happier [34]. When physical exercise is digitized and embedded into social media, it strengthens the existing social network. Research by Gui Xinyu et al. indicates that sharing fitness data with pre-existing social networks motivates users to enhance their existing social relationships [35].

Returning to the research topic of this article, physical exercise and life satisfaction, existing research has shown that physical exercise can improve participant life satisfaction [9,16,36,37,38,39,40,41,42,43]. People who regularly participate in physical exercise have higher life satisfaction than those who do not exercise [44]. Middle-aged adult males who engaged in regular physical exercise reported greater levels of overall life satisfaction [45]. There is a significant positive correlation between the time and frequency of physical exercise and college students’ life satisfaction [46]. In the short term, exercise induces positive mood states, and in the long term, regular exercise leads to greater happiness [47]. However, some studies have also pointed out that the correlation between the two is unstable. Sidney and Shephard concluded that although physical exercise improved the physical condition of the elderly, it failed to improve their life satisfaction [48]. The research by Dai Q. and Yao J.-X. shows that there is no direct correlation between physical exercise and the life satisfaction of the elderly. Physical exercise affects the life satisfaction of elderly people through two mediating variables: friend support and self-efficacy [49]. There are also studies indicating that excessive exercise reduces the life satisfaction of elderly people [50]. Maher et al.’s research found that there is no correlation between physical exercise and life satisfaction in young adulthood [51]. Research has also found that moderate-intensity physical exercise can increase happiness, but high-intensity exercise can reduce happiness [52]. Low-intensity and moderate-intensity physical exercise bring happiness, while the effects of high-intensity exercise cannot be verified [53]. First, there is still no final conclusion regarding the relationship between physical exercise and life satisfaction in existing research. Second, most of the existing research has focused narrowly on the relationship between physical exercise and life satisfaction in specific age groups (elderly, young people, etc.), specific gender groups (such as males), and specific identity groups (such as students), lacking a more comprehensive macro perspective that examines this relationship across all age groups, genders, and identities. Third, among the studies that believe there is a correlation between the two, there is no consensus on how the frequency and duration of physical exercise affect life satisfaction. Finally, there is insufficient discussion on the mechanism of the correlation between the two.

The fitness culture [33] is spreading among the urban population in China, with an increasing number of Chinese people participating in daily physical exercise. According to the National Fitness Survey Bulletin of the General Administration of Sports of China in 2014, the percentage of people who regularly took part in physical exercise in 2014 was 33.9% (including children and adolescents), an increase of 5.7 percentage points compared with 2007, and included 14.7% of people aged 20 and over, of which 19.5% were urban residents, an increase of 48.0% compared with 2007. It can be seen that urban citizens’ physical exercise activities are increasing day by day.

Based on the previous discussion, we propose the following research hypotheses:

**Hypothesis** **1.**
*The more frequent the frequency of physical exercise, the higher the satisfaction of urban residents with their lives.*


**Hypothesis** **2.**
*The longer the duration of physical exercise, the higher the satisfaction of urban residents with their lives.*


**Hypothesis** **3.**
*The frequency of physical exercise is positively associated with life satisfaction among urban residents, mediated through an increase in personal popularity.*


**Hypothesis** **4.**
*The frequency of physical exercise is positively related to life satisfaction among urban residents, mediated through an increase in positive emotions.*


**Hypothesis** **5.**
*The duration of physical exercise is positively associated with life satisfaction among urban residents, mediated by an increase in personal popularity.*


**Hypothesis** **6.**
*The duration of physical exercise is positively correlated with life satisfaction among urban residents, mediated through an enhancement of positive emotions.*


## 2. Methods

### 2.1. Data Source and Analysis Methods

The data used in this study come from the 2018 China Family Panel Studies (CFPS), a panel data collection project managed by Peking University in China. CFPS focuses on the economic and non-economic welfare of Chinese residents, as well as many research topics including economic activities, educational achievements, family relations and family dynamics, population migration, health, etc. It is a national, large-scale, multi-disciplinary social tracking survey project. CFPS samples cover 25 provinces/cities/autonomous regions, and the target sample size is 16,000 households. The respondents include all family members in the sample households. A total of 32,669 valid cases were collected in CFPS 2018, including 7872 urban cases, 22,259 rural cases, and 2538 other cases.

Considering that people in rural areas of China rarely participate in physical exercise, this article only retains urban samples. (Physical exercise is defined as an active and regular physical activity undertaken for the purpose of exercising, excluding passive activities like agricultural labor. While it is often assumed that rural Chinese individuals may not actively engage in physical exercises, we understand this may be a generalization. However, for the sake of clarity and standardization, we have chosen to limit the scope of this article to urban samples, considering the presumed lower participation rate of physical exercise in rural areas.) After handling missing values, the final sample of urban residents was 7423 people. It is worth mentioning that from 2018 to now, CFPS has released the latest data for 2020. Considering that most people affected by the COVID-19 epidemic were isolated at home that year, which seriously affected the results we want to test, the data for 2020 are not used.

We use a multiple linear regression model and StataMP 17 statistical software for data analysis. The set *p*-values are 0.001, 0.01, and 0.05.

### 2.2. Measurements

#### 2.2.1. Dependent Variable

Following the CFPS 2018 survey questionnaire, we used the question, “How satisfied are you with your life?” to measure the life satisfaction of urban residents. Respondents had five options: 1 = not at all satisfied, 2 = not satisfied, 3 = average, 4 = satisfied, 5 = very satisfied.

#### 2.2.2. Independent Variables

To measure the condition of exercise of urban residents, we selected two questions from the 2018 CFPS questionnaire. One question was, “How many times have you exercised in the past week?” and the other was, “How long have you exercised in the past week?”; the former can be answered from 0 to 50 times, while the latter can be answered from 0.1 to 105.0 h, the higher the value, the higher the frequency and intensity of exercise. The exercises here include walking, long-distance running, jogging, and mountaineering; practicing martial arts such as Tai Chi and qigong; indoor and outdoor dance; fitness exercises; aerobics; yoga; various ball games such as small and large balls; water sports such as swimming, diving, rowing, and sailing; winter ice and snow sports; as well as physical contact sports such as wrestling, judo, and boxing.

#### 2.2.3. Mediating Variables

To reveal the mechanism of the impact of physical exercise on life satisfaction, we used popularity and positive emotions as mediating variables, corresponding to “How good the social relationship is” and “The frequency of various feelings or behaviors occurring in the past week” with 8 statements (the 8 statements are the following: “I feel emotionally low”, “I feel that doing anything is difficult”, “My sleep is not good”, “I feel happy”, “I feel lonely”, “I live happily”, “I feel sad”, “I feel that life cannot continue”), respectively. The former has a minimum score of 0 and a maximum score of 10, while the latter has a minimum score of 4 and a maximum score of 32 (each question scores 1–4 points, the more positive the emotion, the higher the score).

#### 2.2.4. Control Variables

To eliminate error caused by the omission of variables from the model estimation to the best extent possible, we introduced other related control variables based on existing research results. The control variables in this article included the following: individual’s gender (female = 0; male = 1), marital status (unmarried = 0; married = 1), age, years of education, self-rated income level in local terms (1–5), and self-rated health (1–5).

Regarding years of education, the options provided in the questionnaire include “illiterate/semi-illiterate”, “elementary school”, “junior high school”, “high school/technical school/vocational high school”, “associate degree”, “undergraduate degree”, “master’s degree”, and “doctoral degree”. In China, in most cases, they correspond to education years of 0, 6, 9, 12, 15, 16, 19, and 23 years, respectively. Currently, most studies have shown that relative deprivation of income has a greater impact on happiness compared to absolute income [4,54]. Therefore, we select self-rated income level in local terms as the control variable to control for economic level.

## 3. Results

### 3.1. Descriptive Statistical Analysis

Table 1 shows the basic situation of the variables in this study. There are 3641 (49.05%) women and 3782 (50.95%) men in the sample, and 1371 (18.47%) samples are unmarried and 6052 (81.53%) samples are married. The average age is 49.55 years old and the average number of years of education is 10.45 years. Respondents rated their income level in local terms and their health status as medium, giving 2.87 points and 2.94 points, respectively, from the full score of 5 points. The average number of physical exercise sessions per week is 3.41 times, and the average length of physical exercise is 4.74 h per week. The average of life satisfaction with a full score of 5 points is 4.01 points. This illustrates that the life satisfaction of Chinese urban residents is on the high side.

### 3.2. Regression Analysis

Table 2 shows the correlation between frequency and duration of exercise and life satisfaction of Chinese urban residents without any control variables. Among them, the *p* value between exercise frequency and life satisfaction is statistically significant, and the *p* value between the duration of exercise and life satisfaction is also statistically significant, which indicates that exercise condition affects the life satisfaction of Chinese urban residents.

We used multiple regressions in Table 3 to reveal whether the relationship between physical exercise and life satisfaction is still significant when the control variables are added. Model 1 introduced control variables into the model, and we drew the following conclusions by analyzing the model fit results: gender, marital status, age, years of education, income level in local terms, and health condition all had a significant relationship with the life satisfaction of urban residents. Specifically, women have higher life satisfaction than men. Being married has a higher level of life satisfaction than not being married. The higher the age, income level in local terms, and health condition, the higher the life satisfaction. It is worth noting that our study found that the higher the number of years of education, the lower the life satisfaction. One possible reason is that longer years of education in China means greater academic burden and more intense competition among peers, so their life satisfaction is not very high. Studies on education level and depression have also confirmed that after reaching a certain level of education, the higher the level of education, the more likely it is to experience depression [55,56,57].

On this basis, we added the first independent variable exercise frequency in Model 2, and we found that there is still a significant correlation between the frequency of physical exercises and life satisfaction (*p* < 0.001); each time the number of physical exercise sessions increases 1 time, the life satisfaction of urban residents will increase 0.0123 points. On the basis of Model 1, we added the second independent variable duration of exercise in Model 3, and the result is still significant; for every 1 h increase in the length of physical exercise, the life satisfaction of urban residents will be increased by 0.00423 points. Hypotheses 1 and 2 are verified.

To verify hypotheses 3, 4, 5, and 6, namely the mediating role of popularity and positive emotions in the impact of physical exercise frequency and duration on life satisfaction, we conducted regression analysis of the mediating variable on the dependent variable and regression analysis of the independent variable on the mediating variable, respectively. The results are presented in Table 4 and Table 5.

Exercise frequency → Popularity and Positive emotions → Life satisfaction. As shown in Model 4, we add the first mediator variable on top of Model 2: popularity. We found a significant correlation between popularity and life satisfaction (*p <* 0.001). For every 1-point increase in popularity, life satisfaction increases by 0.0842 points. As shown in Model 5, based on Model 4, we add a second mediator variable: positive emotions. We found a significant correlation between positive emotions and life satisfaction (*p <* 0.001). For every 1-point increase in positive emotions, life satisfaction increases by 0.0549 points. From Model 2 to Model 4 and then to Model 5, the estimated coefficient of exercise frequency on life satisfaction gradually decreases, from 0.0123 points to 0.0106 points and then to 0.00591 points. Therefore, the two mediating variables of popularity and positive emotions can explain the partial impact of exercise frequency on life satisfaction.

Duration of exercise → Popularity and Positive emotions → Life satisfaction. As shown in Model 6, we add the first mediating variable on top of Model 3: popularity. We conclude that in this model, there is a significant correlation between popularity and life satisfaction (*p <* 0.001). For every 1-point increase in popularity, life satisfaction increases by 0.0845 points. As shown in Model 7, based on Model 6, we add a second mediator variable: positive emotions. We found a significant correlation between positive emotions and life satisfaction (*p <* 0.001). For every 1-point increase in positive emotions, life satisfaction increases by 0.0551 points. From Model 3 to Model 6 and then to Model 7, the estimated coefficient of duration of exercise on life satisfaction gradually decreases, from 0.00423 points to 0.00368 points and then to 0.00235 points. Therefore, the two mediating variables of popularity and positive emotions can explain the partial impact of duration of exercise on life satisfaction.

In order to ultimately confirm whether the two mediating variables of popularity and positive emotions are the mediating mechanisms of the impact of physical exercise on life satisfaction, we further tested the relationship between physical exercise and the two mediating variables, and the results are shown in Table 5.

There is a significant correlation between exercise frequency and popularity (*p <* 0.01). Each time the number of physical exercises increases 1 time, the popularity will increase 0.0206 points. There is a significant correlation between exercise frequency and positive emotions (*p <* 0.001). For every increase in exercise frequency, positive emotions increase by 0.0911 points.

There is a significant correlation between exercise duration and popularity (*p <* 0.1). For every increase in exercise frequency, the popularity is 0.00644 points. There is a significant correlation between exercise frequency and positive emotions (*p <* 0.001). For every increase in exercise frequency, positive emotions increase by 0.0260 points.

It can be seen that physical exercise increases urban residents’ popularity and positive emotional level, thereby enhancing life satisfaction. This proves that popularity and positive emotional level are two important mediating mechanisms for the impact of physical exercise on life satisfaction.

## 4. Conclusions

This study analyzed data from the 2018 China Family Panel Studies to examine the relationship between physical exercise and the life satisfaction of urban residents in China. We found that both the frequency and duration of physical exercise have positive correlations with residents’ life satisfaction. The findings verify the previous research results about the positive relationship between physical exercise and life satisfaction [58]. To further reveal the impact mechanism between physical exercise and life satisfaction, we introduce two mediating variables: popularity and positive emotions. The results show that the frequency of physical exercise affects life satisfaction by influencing popularity and positive emotions. Similarly, the duration of physical exercise affects life satisfaction by influencing popularity and positive emotions. Namely, the higher the frequency and duration of physical exercise, the higher the popularity and positive emotion levels among Chinese urban citizens, resulting in an increased life satisfaction.

Residents’ life satisfaction is not only affected by objective conditions, but also by individual cognitive characteristics. The improvement in people’s life satisfaction not only depends on a good policy, social, and cultural environment but also requires a good habit of physical exercise to improve their physical and mental quality.

To explain the positive relationship towards physical exercise and life satisfaction, one possibility is that physical exercise can strengthen physical quality. Good health can improve people’s life satisfaction, especially the elderly [59]. Another possibility is that physical exercise can improve people’s psychological well-being or reduce the level of anxiety and depression [60,61], and then produce emotional benefits. Obviously, good emotions can improve people’s life satisfaction. The third possibility is that in the context of China, in the middle-aged and elderly groups, the group activities represented by square dance are their main physical exercise methods. In addition to the above two benefits, this kind of physical exercise also has strong social characteristics, and these frequent social activities can increase people’s life satisfaction [62].

Although we can infer through the literature references that participating in physical exercise is the reason for increased life satisfaction, the causal relationship between the two cannot be inferred from the data used in this study, which are cross sectional survey data. One possibility is that people with higher life satisfaction are more willing to participate in physical exercise. This is a limitation of our research. In the future, we look forward to research to supplement the limitation at the data level and determine the causal relationship between the two through data analysis.

## Figures and Tables

**Table 1 behavsci-14-00494-t001:** Variable descriptive statistical analysis (N = 7423).

Variable	Variable Definition	Freq./Mean	Percent (%)/SD
DV			
Life satisfaction (1–5)	Assign values of 1–5 from “very dissatisfied” to “very satisfied”.	4.01	0.89
IV			
Exercise frequency (times/week)	The lowest value is 0, the highest value is 35, among which 2682 are selected as 0, accounting for 36%.	3.41	3.30
Duration of exercise (hour/week)	The lowest value is 0, the highest value is 90, among which 2682 are selected as 0, accounting for 36%.	4.74	6.92
MV			
Popularity (0–10)	Assign values of 0–10 from “very unpopular” to “very popular”.	7.17	1.80
Positive emotions (8–32)	It involves 8 CESD variables, with a minimum value of 1 and a maximum value of 4 for each variable. After adding up, the minimum value is 8 and the maximum value is 32. The higher the score, the higher the level of positive emotions.	27.08	3.76
CV			
Gender			
Female (0)		3641	49.05
Male (1)		3782	50.95
Marital status			
Unmarried (0)		1371	18.47
Married (1)		6052	81.53
Age	The minimum age is 16, and the maximum age is 96.	49.55	16.23
Years of education	The minimum education period is 0 years, and the maximum education period is 23 years. Among them, 722 are illiterate, accounting for 9.73%.	10.45	4.70
Self-rated income level in local terms (1–5)	Assign values of 1–5 from “very low” to “very high”.	2.87	0.99
Self-rated health (1–5)	Assign values of 1–5 from “unhealthy” to “very healthy”.	2.92	1.09

DV: dependent variable, IV: independent variable, MV: mediator variable, CV: control variable.

**Table 2 behavsci-14-00494-t002:** Bi-variate analyses of exercise and life satisfaction.

	Exercise Frequency	Duration of Exercise
Life satisfaction	*r* = 9.53, *p* < 0.001	*r* = 8.79, *p* < 0.001

*r:* Spearman’s rank correlation coefficient.

**Table 3 behavsci-14-00494-t003:** Multiple regressions with life satisfaction as their dependent variables (N = 7423).

Variables	Model 1	Model 2	Model 3
Gender (male)	−0.0540 **	−0.0572 **	−0.0573 **
	(−2.79)	(−2.96)	(−2.96)
Married	0.179 ***	0.182 ***	0.182 ***
	(7.14)	(7.28)	(7.25)
Age	0.00656 ***	0.00564 ***	0.00613 ***
	(8.85)	(7.28)	(8.11)
Years of education	−0.0114 ***	−0.0122 ***	−0.0117 ***
	(−4.67)	(−5.00)	(−4.77)
Self-rated income level in local terms	0.256 ***	0.255 ***	0.255 ***
	(25.84)	(25.79)	(25.81)
Self-rated health	0.123 ***	0.120 ***	0.121 ***
	(13.19)	(12.90)	(13.01)
Exercise frequency		0.0123 ***	
		(4.05)	
Duration of exercise			0.00423 **
			(2.98)
F	205.94	179.23	177.98
*p*	<0.001	<0.001	<0.001
R^2^	0.1428	0.1447	0.1438

** *p* < 0.01, *** *p* < 0.001.

**Table 4 behavsci-14-00494-t004:** Regression estimation results of mediating variables on life satisfaction (N = 7423).

Variables	Model 4	Model 5	Model 6	Model 7
Gender (male)	−0.0481 *	−0.0769 ***	−0.0482 *	−0.0772 ***
	(−2.53)	(−4.14)	(−2.53)	(−4.16)
Married	0.182 ***	0.128 ***	0.182 ***	0.128 ***
	(7.40)	(5.31)	(7.38)	(5.30)
Age	0.00453 ***	0.00305 ***	0.00493 ***	0.00324 ***
	(5.91)	(4.08)	(6.60)	(4.43)
Years of education	−0.0141 ***	−0.0179 ***	−0.0136 ***	−0.0177 ***
	(−5.85)	(−7.62)	(−5.66)	(−7.53)
Self-rated income level in local terms	0.233 ***	0.217 ***	0.233 ***	0.217 ***
	(23.67)	(22.66)	(23.68)	(22.66)
Self-rated health	0.101 ***	0.0484 ***	0.102 ***	0.0485 ***
	(10.93)	(5.18)	(11.01)	(5.19)
Exercise frequency	0.0106 ***	0.00591 *		
	(3.53)	(2.02)		
Popularity	0.0842 ***	0.0694 ***	0.0845 ***	0.0696 ***
	(15.67)	(13.18)	(15.73)	(13.20)
Positive Emotions		0.0549 ***		0.0551 ***
		(20.76)		(20.87)
Duration of exercise			0.00368 **	0.00235
			(2.64)	(1.73)
F	192.67	229.06	191.84	228.91
*p*	<0.001	<0.001	<0.001	<0.001
R^2^	0.1721	0.2176	0.1715	0.2175

* *p* < 0.05, ** *p* < 0.01, *** *p* < 0.001.

**Table 5 behavsci-14-00494-t005:** Regression estimation results of the relationship between exercise and mediating variables (N = 7423).

Variables	Popularity	Positive Emotions	Popularity	Positive Emotions
Gender (male)	−0.108 **	0.495 ***	−0.107 **	0.498 ***
	(−2.62)	(6.03)	(−2.61)	(6.06)
Married	−0.00107	0.988 ***	−0.00234	0.980 ***
	(−0.02)	(9.30)	(−0.04)	(9.21)
Age	0.0133 ***	0.0305 ***	0.0141 ***	0.0346 ***
	(8.05)	(9.27)	(8.80)	(10.77)
Years of education	0.0223 ***	0.0752 ***	0.0233 ***	0.0799 ***
	(4.29)	(7.24)	(4.49)	(7.69)
Self-rated income level in local terms	0.265 ***	0.352 ***	0.265 ***	0.355 ***
	(12.60)	(8.40)	(12.62)	(8.45)
Self-rated health	0.228 ***	1.022 ***	0.229 ***	1.031 ***
	(11.47)	(25.84)	(11.57)	(26.03)
Exercise frequency	0.0206 **	0.0911 ***		
	(3.17)	(7.05)		
Duration of exercise			0.00644 *	0.0260 ***
			(2.13)	(4.31)
F	61.45	170.73	60.61	165.61
*p*	<0.001	<0.001	<0.001	<0.001
R^2^	0.0548	0.1388	0.0541	0.1352

* *p* < 0.05, ** *p* < 0.01, *** *p* < 0.001.

## Data Availability

The data that support the findings of this study are available from Institute of Social Science Survey, Peking University. Data are available from Peking University Research Data Platform https://opendata.pku.edu.cn/dataset.xhtml?persistentId=doi:10.18170/DVN/45LCSO (accessed on 13 May 2023) with the permission of Institute of Social Science Survey, Peking University.

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
