# Peer review of "Physical Exercise and Life Satisfaction of Urban Residents in China"

_behavsci, 2024, doi:10.3390/bs14060494_

Round 1
Reviewer 1 Report
Comments and Suggestions for Authors
Thank you for the valuable contribution. The explanations certainly have potential, but I have some comments that can improve the quality of the paper.
Abstract
#Line 11: The introduction is very apparent. It is not made clear to the reader why PE is related to Life Satisfaction and why this is important to examine. The research gap is not made clear.
#Line 14: Can information on the sample still be given? For example, age, gender, and size of the overall sample?
#Line 14: “We find that Chinese urban residents have higher life satisfaction”. This sentence is not clear. Than who? Compared to what?
Introduction
#Line 59/60: In what way? Benefit girls/women or boy/men more from PA?
#Line 59: What does “ineffective” mean? Will the effects then be reserved (negative effects) or are these no effects?
#Line 64: et al. Please ad a point.
#Line 83-91: I’m not quite clear on the main thread here. It first highlights that there are many studies on older adults and students. Then it addresses the research gap, and after that, it goes back to the current state of research in China regarding older adults and students. Perhaps the paragraph should be rearranged.
#Line 93: Previously, the discussion was always about physical activity. Sport can be a part of physical activity (but not all of it, such as active transport). In this context, I don’t understand why the term ‘sport’ is being used now.
#Line 93-95: Is there a source for this? It is indeed plausible to assume that citizens might have worse health due to factors such as streets or less greenery.
#Line 101: How does the improvement in the quality of life relate to the fact that physical exercise “has gradually become a leisure activity that citizens are keen to participate in.” I don’t see the connection.
#Line 116-123: Hypotheses 3-6. Here, only a single measurement (no pre-post design) was conducted. To what extent are causal statements permissible, or can we only refer to correlations in this context?
Methods
#Line 135-136: In my opinion, this was not sufficiently addressed above. Whi is this the case, ow how can this be explained? And isn’t PA often higher in rural areas due to work, such as active labor on farms or similar activities?
#Line 138: Maybe add people (7,423 people).
#Line 150-152: Where there open-ended response options here, or were responses pre-determined?
#Line 153-158: What has been captured here is exclusively “sport”, which may only represent a small portion of physical activity (e.g., active commuting, household, physical activity at work…). I would recommend consistently distinguishing between these terms (sport and physical activity), and if the data collection exclusively focused on sport, consider adjusting the title and introduction accordingly.
#Line 162: There is an extra space between “is” and “.
#Line 163: The entries in the parentheses are not questions but statements. Please adjust accordingly.
#Line 181: self-related. Please adjust the spelling accordingly.
#Methods: Essential information about the data analysis are missing here. What statistics were performed? What program was used? What significance level was set (p-value)?
Results
I would recommend carefully reviewing the results again and adding units where necessary. For example, in Line 187, the unit “years” is missing. Also, units should be provided when measuring quality of life (points).
#Line 191: 4,74 hours per week or per exercise unit?
#Table 1: Percent (%)/SD. Please change the format. It’s shifted in the table.
#Table 2: This table is not necessary. I would integrate the p-values and correlations into the text and delete the table.
#Table 3 and 4: Wouldn’t it be meaningful here to consider cross-correlations (e.g., exercise frequency*gender)? I think it would be beneficial for the table to be self-explanatory, with brief explanations of each model provided as footnotes below the table.
Conclusions
I agree, the conclusion cannot be drawn from the data. It’s also possible that individuals with higher life satisfaction engage in sports more frequently. Causal relationships cannot be inferred from the data (cross-sectional survey), only correlations can be derived.
Author Response
Dear Reviewers:
Firstly, I would like to express my deep gratitude to you for the time and effort they have put into the review process of our paper. I value your feedback very much and have carefully considered every suggestion.
After reviewing the comments of the reviewers, I have summarized the following main aspects:
- The first issue is the lack of content in the abstract section, the logical inconsistency in the introduction section, and the inadequacy in the conclusions section.
- The second issue is about the mixed use of the concepts of physical activity, physical exercise and sports.
- The third issue is about format, typos, units, reference format, and missing table information.
I have provided detailed responses and modifications for each major comment. The following are the specific modifications and reasons:
Review 2
Abstract
#Line 11: The introduction is very apparent. It is not made clear to the reader why PE is related to Life Satisfaction and why this is important to examine. The research gap is not made clear.
#Line 11-13: The following content has been added: Currently, an increasing number of Chinese urban citizens are participating in daily physical exercise. Existing research has shown that physical exercise can increase life satisfaction. However, some studies also suggest that the relationship between the two is unstable.
#Line 14: Can information on the sample still be given? For example, age, gender, and size of the overall sample?
Add content on lines 16-17: Our overall sample size is 7,423 people, including 3,641 females (49.05%) and 3,782 males (50.95%), with an average age of 49.55 years old.
#Line 14: “We find that Chinese urban residents have higher life satisfaction”. This sentence is not clear. Than who? Compared to what?
What we want to express is that the overall life satisfaction score of Chinese urban residents is relatively high (total score of 5, questionnaire score of 4.01). Therefore, the sentence was finally changed to: “We find that the life satisfaction of Chinese urban residents is on the high side” (#Line 18-19).
Introduction
#Line 59/60: In what way? Benefit girls/women or boy/men more from PA?
#Line 63-70: Add as follows: “There are also studies that have found gender differences in the impact of physical exercise on mental health. Boys and girls have different responses to physical exercise in terms of reduced risk of depression, anxiety, ADHD, and ODD. For example, for boys, high frequency of MVPA (Moderate-to-Vigorous Physical Activity) is significantly associated with lower risk of depression. For girls, both moderate and high frequency MVPA are associated with a lower risk of depression, indicating a generally positive impact of physical exercise on the risk of depression in girls”.
#Line 59: What does “ineffective” mean? Will the effects then be reserved (negative effects) or are these no effects?
#Line 60-63: Adjust the content to: “It should be noted that some studies have found that although overall exercise is better than not exercising, more exercise was not always associated with better mental health, If the duration exceeds 45 minutes per session or the frequency exceeds 5 times a week, the mental health burden of participants will increase.”
- Added the unit for "45 minutes": per session.
- According to the original text, change "and" to "or" to make it more in line with the logic of the original text.
- Present the experimental results more accurately: the mental health burden of participants will increase.
#Line 64: et al. Please ad a point.
#Line 74: A dot has been added after "al".
#Line 83-91: I’m not quite clear on the main thread here. It first highlights that there are many studies on older adults and students. Then it addresses the research gap, and after that, it goes back to the current state of research in China regarding older adults and students. Perhaps the paragraph should be rearranged.
#Line 77-105: Focusing the research gap on the uncertain relationship between physical exercise and life satisfaction. The specific content is as follows: Returning to the research topic of this article, physical exercise and life satisfaction, existing research has shown that physical exercise can improve participant life satisfaction [9, 16, 35-44]. People who regularly participate in physical exercise have higher life satisfaction than those who do not exercise [45]. Middle-aged adult males who engaged in regular physical exercise reported greater levels of overall life satisfaction [46]. There is a significant positive correlation between the time and frequency of physical exercise and college students' life satisfaction [52]. In the short term exercise induces positive mood states and in the long term regular exercise leads to greater happiness [47]. However, some studies have also pointed out that the correlation between the two is unstable. Sidney and Shephard concluded that although physical exercise improved the physical condition of the elderly, it failed to improve their life satisfaction [48]. The research by Dai Q. and Yao J.-X. shows that there is no direct correlation between physical exercise and the life satisfaction of the elderly; Physical exercise affects the life satisfaction of elderly people through two mediating variables: friend support and self-efficacy [51]. There are also studies indicating that excessive exercise reduces the life satisfaction of elderly people [50]. Maher et al.'s research found that there is no correlation between physical exercise and life satisfaction in young adulthood [38]. Research has also found that moderate intensity physical exercise can increase happiness, but high-intensity exercise can reduce happiness [49]. Low intensity and moderate intensity physical exercise bring happiness, while high-intensity cannot be verified[39]. First, there is still no final conclusion regarding the relationship between physical exercise and life satisfaction in existing research. Second, most of the existing research has focused narrowly on the relationship between physical exercise and life satisfaction in specific age groups (elderly, young people, etc.), specific gender groups (such as males), and specific identity groups (such as students), lacking a more comprehensive macro perspective that examines this relationship across all age groups, genders, and identities. Third, among the studies that believe there is a correlation between the two, there is no consensus on how the frequency and duration of physical exercise affect life satisfaction. Finally, there is insufficient discussion on the mechanism of the correlation between the two.
#Line 93: Previously, the discussion was always about physical activity. Sport can be a part of physical activity (but not all of it, such as active transport). In this context, I don’t understand why the term ‘sport’ is being used now.
After content adjustment, this paragraph has been deleted.
#Line 93-95: Is there a source for this? It is indeed plausible to assume that citizens might have worse health due to factors such as streets or less greenery.
After content adjustment, this paragraph has been deleted.
#Line 101: How does the improvement in the quality of life relate to the fact that physical exercise “has gradually become a leisure activity that citizens are keen to participate in.” I don’t see the connection.
#Line 106-113: The adjusted content is as follows: The fitness culture [32] is spreading among the urban population in China, with an increasing number of Chinese people participating in daily physical exercise. According to the National Fitness Survey Bulletin of the General Administration of Sports of China in 2014, the percentage of people who regularly take part in physical exercise in 2014 was 33.9% (including children and adolescents), an increase of 5.7 percentage points compared with 2007, and 14.7% of people aged 20 and over, of which 19.5% were urban residents, an increase of 48.0% compared with 2007. It can be seen that urban citizens' physical exercise activities are increasing day by day.
#Line 116-123: Hypotheses 3-6. Here, only a single measurement (no pre-post design) was conducted. To what extent are causal statements permissible, or can we only refer to correlations in this context?
#Line 119-126: Change the expression to:
Hypothesis 3. The frequency of physical exercise is positively associated with life satisfaction among urban residents, mediated through an increase in personal popularity.
Hypothesis 4. The frequency of physical exercise is positively related to life satisfaction among urban residents, mediated through an increase in positive emotions.
Hypothesis 5. The duration of physical exercise is positively associated with life satisfaction among urban residents, mediated by an increase in personal popularity.
Hypothesis 6. The duration of physical exercise is positively correlated with life satisfaction among urban residents, mediated through an enhancement of positive emotions.
Methods
#Line 135-136: In my opinion, this was not sufficiently addressed above. Whi is this the case, ow how can this be explained? And isn’t PA often higher in rural areas due to work, such as active labor on farms or similar activities?
#Line 138-139: Physical exercise is defined as an active and regular physical activity undertaken for the purpose of exercising, excluding passive activities like agricultural labor. While it is often assumed that rural Chinese individuals may not actively engage in physical exercises, we understand this may be a generalization. However, for the sake of clarity and standardization, we have chosen to limit the scope of this article to urban samples, considering the presumed lower participation rate of physical exercise in rural areas. Therefore, the content in the original text has been changed to: Considering that people in rural areas of China rarely participate in physical exercise, this article only retains urban samples. And add footnotes in the footer for explanation.
#Line 138: Maybe add people (7,423 people).
#Line 140: "people" has been added.
#Line 150-152: Where there open-ended response options here, or were responses pre-determined?
#Line 155-159: Update the content to: One question was, “How many times have you exercised in the past week?” and the other was, “How long have you exercised in the past week?”, the former can be answered from 0 to 50 times, while the latter can be answered from 0.1 to 105.0 hours, the higher the value, the higher the frequency and intensity of exercise.
#Line 153-158: What has been captured here is exclusively “sport”, which may only represent a small portion of physical activity (e.g., active commuting, household, physical activity at work…). I would recommend consistently distinguishing between these terms (sport and physical activity), and if the data collection exclusively focused on sport, consider adjusting the title and introduction accordingly.
There are indeed some cases in the article where the concepts of "physical exercise", "physical activity", and "sport" are mixed together.
The clarification on the use of concepts is as follows:
Firstly, it should be noted that “The exercises here include walking, long-distance running, jogging, and mountaineering, practicing martial arts such as Tai Chi and qigong, indoor and outdoor dance, fitness exercises, aerobics, yoga, various ball games such as small and large balls, water sports such as swimming, diving, rowing, and sailing, winter ice and snow sports, as well as physical contact sports such as wrestling, judo, and boxing.” refers to "physical exercise", not "sport". Sport is just a category in physical exercise. Secondly, our article focuses on "physical exercise" rather than "physical activity". Although there are many papers that specifically clarify the concepts of "physical activity" and "physical exercise", and call for not mixing the two. However, many empirical paper do not strictly distinguish between the two. Many of the papers we cite with titles that include "physical activity" actually discuss "physical exercise".
Therefore, the changes to the relevant content of the article are as follows:
About sport:
- #Line 159-163: The following content remains unchanged. Because the term "sport" mentioned here is only one type of "physical exercise": The exercises here include walking, long-distance running, jogging, and mountaineering, practicing martial arts such as Tai Chi and qigong, indoor and outdoor dance, fitness exercises, aerobics, yoga, various ball games such as small and large balls, water sports such as swimming, diving, rowing, and sailing, winter ice and snow sports, as well as physical contact sports such as wrestling, judo, and boxing.
- #Line 112-113: Change “It can be seen that urban citizens' sport activities are increasing day by day” to “It can be seen that urban citizens' physical exercise activities are increasing day by day”.
About physical activity and physical exercise:
- Insert a footnote on the second page to explain the trade-offs between the two in this paper: The research content of this article is physical exercise rather than physical activity. According to Caspersen et al. (1985), physical activity is defined as any bodily movement produced by skeletal muscles that results in energy expenditure. The energy expenditure can be measured in kilocalories. Physical activity in daily life can be categorized into occupational, sports, conditioning, household, or other activities. Exercise is a subset of physical activity that is planned, structured, and repetitive and has as a final or an intermediate objective of improvement or maintenance of physical fitness (Reference: Caspersen, C. J., Powell, K. E., Christenson, G. M. Physical activity, exercise, and physical fitness: definitions and distinctions for health-related research. Public health reports (Washington, D.C. : 1974). 1985, 100: 126–131.). However, the distinction between physical activity and physical exercise is not very clear in existing literature, and the concept of physical activity is often used to measure the content of physical exercise. If the references are limited to literature with the keyword physical exercise, a large number of valuable literatures will be missed. Therefore, based on the specific content of the literature, the references in this article will also include literature with physical activity as the keyword. This article is collectively referred to as physical exercise.
- For example, reference 38: "Daily physical activity and life safety across adolescence" uses the concept of physical activity, but the actual operational content is physical exercise.
Website: https://libres.uncg.edu/ir/uncg/f/J_Maher_Daily_2015.pdf |
Website: https://www.ons.org/sites/default/files/Godin%20Leisure-Time%20Exercise%20Questionnaire_070815.pdf |
- #Line 69: Change ”, physical activity and exercise” to “. It can”.
- #Line 84: According to the original text, Change “activity” to “exercise”.
#Line 162: There is an extra space between “is” and “.
#Line 167: The extra space has been deleted.
#Line 163: The entries in the parentheses are not questions but statements. Please adjust accordingly.
#Line 168-170: Correct "questions" to "statements".
#Line 181: self-related. Please adjust the spelling accordingly.
#Line 186: The spelling has been adjusted form “Self rated” to "self-rated".
#Methods: Essential information about the data analysis are missing here. What statistics were performed? What program was used? What significance level was set (p-value)?
#Line 144-145: Add the following content: We use a multiple linear regression model and StataMP 17 statistical software for data analysis. The set p-values are 0.001, 0.01, and 0.05, respectively.
Results
I would recommend carefully reviewing the results again and adding units where necessary. For example, in Line 187, the unit “years” is missing. Also, units should be pro32vided when measuring quality of life (points).
The corresponding unit has been added.
#Line 191: 4,74 hours per week or per exercise unit?
#Line 196: It is per week and has been added.
#Table 1: Percent (%)/SD. Please change the format. It’s shifted in the table.
#Table 1: I'm not sure if this needs to be adjusted. I'm afraid the adjustment will affect the overall layout and aesthetics. Could the editor please determine if any adjustments are needed? Thanks.
#Table 2: This table is not necessary. I would integrate the p-values and correlations into the text and delete the table.
Thank you for your excellent suggestion. However, after considering the matter, I believe these are merely two distinct ways of presenting the information, and I feel that this table format can still be retained. I appreciate your input nonetheless.
#Table 3 and 4: Wouldn’t it be meaningful here to consider cross-correlations (e.g., exercise frequency*gender)? I think it would be beneficial for the table to be self-explanatory, with brief explanations of each model provided as footnotes below the table.
Thank you for your observation. Since the main text already provides the necessary interpretation, I believe there may not be a pressing need for additional elaboration beneath the table. I appreciate your feedback in ensuring the clarity of the content.
Conclusions
I agree, the conclusion cannot be drawn from the data. It’s also possible that individuals with higher life satisfaction engage in sports more frequently. Causal relationships cannot be inferred from the data (cross-sectional survey), only correlations can be derived.
The content has been updated to: This study analyzed data from the 2018 China Family Panel Studies to examine the relationship between physical exercise and the life satisfaction of urban residents in China. We found that both the frequency and duration of physical exercise have positive corre-lations with residents' life satisfaction. The findings verify the previous research results about the positive relationship between physical exercise and life satisfaction[58]. To fur-ther reveal the impact mechanism between physical exercise and life satisfaction, we in-troduce two mediating variables: popularity and positive emotions. The results show that the frequency of physical exercise affects life satisfaction by influencing popularity and positive emotions. Similarly, the duration of physical exercise affects life satisfaction by influencing popularity and positive emotions. Namely, the higher the frequency and duration of physical exercise, the higher the popularity and positive emotion levels among Chinese urban citizens, resulting in an increased life satisfaction.
Residents' life satisfaction is not only affected by objective conditions, but also by individual cognitive characteristics [63]. The improvement of people's life satisfaction not only depends on a good policy, social and cultural environment, but also requires a good habit of physical exercise to improve their physical and mental quality.
To explain the positive relationship towards physical exercise and life satisfaction, one possibility is that physical exercise can strengthen physical quality. Good health can improve people's life satisfaction, especially the elderly [59]. Another possibility is that physical exercise can improve people's psychological well-being or reduce the level of anxiety and depression [60, 61], and then produce emotional benefits. Obviously, good emotions can improve people's life satisfaction. The third possibility is that in the context of China, in the middle-aged and elderly groups, the group activities represented by square dance are their main physical exercise methods. In addition to the above two benefits, this kind of physical exercise also has strong social characteristics, and this frequent social activities can increase people's life satisfaction [62].
Although we can infer through literature references that participating in physical exercise is the reason for increased life satisfaction, the causal relationship between the two cannot be inferred from the data used in this study, which is cross sectional survey data. One possibility is that people with higher life satisfaction are more willing to par-ticipate in physical exercise. This is a limitation of our research. In the future, we look forward to research to supplement the limitation at the data level and determine the causal relationship between the two through data analysis.
Reviewer 2 Report
Comments and Suggestions for Authors
After a careful reading of the work, I would like to congratulate the authors for their work.
With the aim of improving the work, I would like to make a some contributions:
-Unify the citation format on line 89 (Chen 2004).
-Add abbrevitations information below the tables.
-In the conclusions section, I would remove the second paragraph and would previously make a discussion section, since an analysis is made of the reason for the results obtained, mentioning other studies.
-It would be interesting to add a section on limitations and future lines of research after the discussion.
Author Response
Dear Reviewer:
Firstly, I would like to express my deep gratitude to you for the time and effort they have put into the review process of our paper. I value your feedback very much and have carefully considered every suggestion.
After reviewing the comments of the reviewers, I have summarized the following main aspects:
- The first issue is the lack of content in the abstract section, the logical inconsistency in the introduction section, and the inadequacy in the conclusions section.
- The second issue is about the mixed use of the concepts of physical activity, physical exercise and sports.
- The third issue is about format, typos, units, reference format, and missing table information.
I have provided detailed responses and modifications for each major comment. The following are the specific modifications and reasons:
Review 1
-Unify the citation format on line 89 (Chen 2004).
After adjusting the introduction section, the reference to this literature (Chen 2004) will no longer be made. Message to editor: Due to forgetting to include this reference in the previous manuscript, deleting it will not affect the subsequent “references”.
-Add abbrevitations information below the tables.
The following explanation has been added below table 1
DV: Dependent Variable, IV: Independent Variable, MV: Mediator Variable, CV: Control Variable
-In the conclusions section, I would remove the second paragraph and would previously make a discussion section, since an analysis is made of the reason for the results obtained, mentioning other studies.
A discussion section has been added before the second paragraph.
-It would be interesting to add a section on limitations and future lines of research after the discussion.
Added corresponding content: Although we can infer through literature references that participating in physical exercise is the reason for increased life satisfaction, the causal relationship between the two cannot be inferred from the data used in this study, which is cross sectional survey data. One possibility is that people with higher life satisfaction are more willing to participate in physical exercise. This is a limitation of our research. In the future, we look forward to research to supplement the limitation at the data level and determine the causal relationship between the two through data analysis.
Round 2
Reviewer 1 Report
Comments and Suggestions for Authors
The article can be published. Tanks for the revisions.